# How to Select the Leader in a One-Shot Public Goods Game: Evidence from the Laboratory

**DOI:** 10.3390/bs15040444

**Published:** 2025-03-31

**Authors:** Shuo Xu, Wenhao Zhang, Jie Zheng

**Affiliations:** 1International Business School, Beijing Foreign Studies University, Beijing 100089, China; 2Center for Economic Research, Shandong University, Jinan 250100, China; 3Economic Science and Policy Experimental Laboratory, Tsinghua University, Beijing 100084, China

**Keywords:** public goods, cooperation, leadership, laboratory experiment, signaling

## Abstract

We experimentally study how leadership selection mechanisms affect public goods provision. Introducing leadership does not raise contribution. Voluntary leadership performs the worst, primarily because the absence of leadership signals uncooperative play, and candidates free-ride on other leaders. Voluntary leadership from a randomly selected candidate is a promising endogenous leadership selection mechanism, primarily because assuming leadership by revealed preference signals cooperative play, the absence of leadership leaves the possibility of unlucky cooperative candidates, and sole leadership removes the leader’s free-riding incentives.

## 1. Introduction

Many real-world challenges, such as funding public infrastructure, addressing climate change, or maintaining community resources, ask individuals to balance personal interest and collective welfare. A persistent issue in these public goods games is the free-rider problem, which often leads to suboptimal outcomes, where the public good is underprovided, and collective welfare suffers.

Researchers have explored how introducing leadership can enhance public goods provision ([14]; [26]). Leaders can influence group behavior by setting an example ([25]; [33]), coordinating behaviors ([12]; [13]), or signaling commitment ([1]; [21]). However, the effectiveness of leadership depends not just on the presence of a leader but, crucially, on how leaders are selected and the incentives they face. Multiple leaders differ in willingness to cooperate and dilute accountability. The absence of a leader generated from a mechanism with a low barrier strongly suggests pessimism to followers.

This study conceptualizes leadership as the role of the first mover(s) in a sequential public goods game. We assess the effectiveness of four leadership selection mechanisms in comparison to the standard voluntary contribution mechanism in facilitating public goods provision: Random Leadership (*RL*), where a leader is randomly selected; Voluntary Leadership (*VL*), where any player who volunteered becomes a leader; and two hybrid mechanisms, Random Leadership from Voluntary Candidates (*RL-VC*) and Voluntary Leadership from a Random Candidate (*VL-RC*), which combine elements of voluntary and random selection. These mechanisms allow us to explore how different approaches to leadership influence group contributions and cooperation.

Cooperation in public goods experiments with leaders was explained in the literature by behavioral theories such as conditional cooperators ([11]; [12]), reciprocity ([2]; [31]), social norms ([7]; [13]), signaling ([19]; [27]), altruism ([28]), and legitimacy ([3]; [4]). We believe that the *informativeness* of signaling and reciprocity theoretically differentiates the effectiveness of four leadership selection mechanisms. Leadership selection mechanisms function as signaling devices as follows: leaders induce reciprocity by signaling generosity (or lack thereof) through contribution levels.[note 1] Upon observing the contribution levels, followers update their beliefs about their group members’ average degree of generosity. Reciprocity then determines the degree to which the followers cooperate.

We conjecture that Voluntary Leadership from a Random Candidate (*VL-RC*) induces the greatest public goods provision, and a more informative “signal” generating process does not necessarily make a mechanism a better one. Random Leadership (*RL*) *forces* a randomly selected member to be the leader. It is normal to expect a randomly selected member to be selfish or generous. *RL* is the least informative mechanism. In Voluntary Leadership (*VL*), *any* member who *volunteers* becomes a leader. The act of becoming a candidate makes followers believe that upon observing positive contributions, the leader is more likely to be generous. The key downside is that upon observing a less-than-expected contribution level, it is more likely that the leader is a selfish person. The effect is augmented by reciprocity as follows: people reward cooperative behavior and punish free-riding. In Random Leadership from Voluntary Candidates (*RL-VC*), any member who volunteers *must be randomly selected* to be the leader. RL-VC is a weakened version of VL, as follows: followers only observe the contribution level of a *single* leader rather than contributions of *multiple* leaders. The belief on average generosity changes to a lesser degree in RL-VC compared to VL. Voluntary Leadership from a Random Candidate (*VL-RC*) *forces* a randomly selected member to be the candidate, and the *candidate decides* whether to be the leader or not. A positive contribution level strongly signals a generous person, and a negative signal can be explained by bad luck—the member is *forced* to be the candidate in the first place.

Based on our hypothesis, we tested leadership selection mechanisms in the lab and arrived at four main results. First, introducing leadership does not increase average contributions compared with the baseline treatment (*Control*). The *VL* mechanism is the least effective, with contributions significantly lower than in *Control* and other leadership treatments. This result suggests that introducing leadership without addressing the barriers or incentives for leaders to influence group behavior may not enhance cooperation in public goods settings.

Second, the presence of a leader significantly increases group contributions. Groups with leaders consistently contributed more than those without, highlighting the importance of leadership as a signaling device for cooperation. However, the barrier to becoming a leader amplifies the effect of the signal. Mechanisms like *VL-RC*, which restrict leadership to a single and randomly selected candidate, outperform those allowing multiple voluntary candidates (*VL* and *RL-VC*). This indicates that a higher barrier to leadership can foster greater contributions by reducing free-riding and enhancing the perceived credibility of the leader.

Third, leaders in all treatments contributed more than the followers, demonstrating a lead-by-example effect. However, the degree of influence leaders have on followers varies between the mechanisms. The correlation between the leader and follower contributions was positive and significant in *VL* and in *RL-VC*, while there was weaker or no significant correlation in other treatments. A clear path of influence channel may not always be a good thing; in *VL*, the presence of multiple leaders strongly signals cooperation from the leaders, but the overall average contribution is the lowest. The difficulty in inferring followers’ cooperative intentions may dilute the leader’s influence and help conceal the true inclination to cooperation.

Finally, endogenous leadership mechanisms (*VL* and *RL-VC*) that allow multiple candidates tend to induce more selfish leaders than mechanisms with a single candidate (e.g., *VL-RC*). A typical bystander effect ([8]) does not explain the difference in the probability of becoming a leader between *VL* and *RL-VC*. It turns out that there is another layer of *conditional* free-riding incentive in *VL* as follows: the existence of multiple candidates incentivizes candidates to free-ride and not to lead by example. In *RL-VC*, becoming a candidate means that only one subject is eligible to become a leader. When there are multiple leaders, followers in these treatments often base their contributions on the least cooperative leader, suggesting that the quality of leadership, rather than the quantity, drives group outcomes.

The rest of the paper is organized as follows. Section 2 reviews the literature. Section 3 describes the experimental design, Section 4 presents the results, and Section 5 concludes the paper. We attach the experimental instructions in Appendix A. Appendix B includes supplemental analysis and results.

## 2. Literature Review

Research on leadership in the economics literature is pioneered by [19] ([19]). Using a game-theoretical approach, [19] ([19]) focuses on what a leader does to induce a following, either by signaling through information or through costly first-mover actions. [24] ([24]) experimentally tests two forms of leadership in [19] ([19]), as follows: leading by example and leading by sacrifice. In leading by example, the leader makes the decision first, and then, the rest follow. [24] ([24]) finds that leaders expect reciprocity and that pure signaling does not work. In leading by sacrifice, the leader can choose to give up a part of their payoff. The results show that sacrifice functions only when it is lost for the follower. Using a lab experiment, [27] ([27]) shows that leading by example works through signaling rather than reciprocity, by comparing contributions when the return from contributions is privately known to the leader with the scenario when the information is publicly known.

The literature moves on to investigate how the leadership selection mechanisms affect leadership performance. [15] ([15]) considers the performance of a rotated exogenous leadership mechanism with exclusion power. The role of the leader is either fixed throughout the experiment or rotated among the group members. In the other dimension, the leader may or may not be empowered with the right to exclude some group members from the next-round play. Results show that the exclusion power is remarkably effective in promoting public goods contributions, while whether the leader is fixed or rotated makes no significant difference. [29] ([29]) finds that group contributions are closer to the socially optimal level under voluntary leadership than random selection leadership. However, such evidence is not found in [2] ([2]), who use a similar experimental design. [16] ([16]) finds that groups with voluntary leaders outperform groups with involuntary leaders, regardless of whether the leader contributes before or after the other group members do. Leaders can also be chosen via specific voting rules ([15]; [22]). Using a field experiment approach, [20] ([20]) emphasizes the legitimacy of leadership and finds that authority induces greater contribution than randomly selected leadership. [3] ([3]) propose a partly random selection mechanism that combines competitive selections with lotteries. The novel mechanism succeeds with a higher quality of relationships between leaders and followers.

Another stream of the literature considers what types of leaders induce more group contributions. [21] ([21]) finds that leaders of higher social status attract more donations. [28] ([28]) finds conditional cooperators are more likely to volunteer to be leaders. [5] ([5]) finds that groups with appointed leaders who are competent and hardworking outperform groups with self-selected leaders.

The evidence from the reviewed literature regarding the assessments of exogenous versus endogenous leadership is pretty mixed. More importantly, systematic comparisons between different real-world-based leader selection mechanisms are limited. The leadership selection mechanisms studied in this paper are motivated by the observations in charitable fundraising. Some celebrities spontaneously and simultaneously publicly contribute to charities to spur additional donations, establishing voluntary leadership *VL*. However, in some cases, the procedure is set exogenously by a third party, such as a fundraiser, who considers announcing the initial contributions to those who follow ([32]). This mechanism is closer to *RL* and *RL-VC*. More often, the fundraiser can send requests to potential donors but still preserve their right to refuse, thus corresponding to the *VL-RC* mechanism. Some other examples include the execution of global climate policies where one or multiple nations may decide to be the first to take an environmentally progressive action (*VL* or *RL-VC*) or that the superior authority can impose real requirements on a nation first, either in a hard or soft way (*RL* or *VL-RC*, [6]).

We contribute to the literature by providing a comprehensive comparison between exogenous leadership and endogenous leadership, focusing on how the selection procedure shapes the incentives of group members. [2] ([2]) test the differences between randomly selected leadership (*RL*) and randomly selected leadership from voluntary candidates (*RL-VC*). [7] ([7]) implemented a similar method of installing a leader to the proposed *VL-RC* mechanism. However, there are two differences as follows: First, in her design, the installation way is not treated as a specific leader selection mechanism comparable to the random selection mechanism, but rather a rule to divide the groups between groups without a leader (treatment *En-Base*) and groups with a leader leading by example (treatment *En-Lead*) or leading by word (treatment *En-Pledge*) so that the comparison is made at the decision level of the randomly selected candidate, while out analysis makes the comparison at the mechanism level. Second, in [7] ([7]), a subject becomes the leader once and for all periods, resulting in a quite low incidence rate of voluntary leadership, while we allow a new leader to be chosen in each period.

Our study complements [18] ([18]). The paper uses a random-matching fixed-pairing design, while we use a random-matching random-pairing design. The random-pairing design induces a key difference. Unlike in fixed-pairing design, subjects in our design have no history of play to condition on. Thus, subjects are more directed to focus on the one-shot game structure, and the explanation for the results avoids the multiple equilibria concerns typical in repeated games with incomplete information. For instance, we only found that *VL-RC* outperforms *VL* and did not find that *VL-RC* also outperforms *RL* and *RL-VC*, which was the finding of [18] ([18]).

In addition, we give a complete characterization of the differences among the four leadership selection mechanisms, including the attributes and effects regarding the willingness to lead by group members under different scenarios, which gives the foundations for our hypotheses and predicts that *VL* performs the worst and *VL-RC* performs the best within the endogenous leadership selection mechanism considered.

## 3. Experimental Design

### 3.1. Treatments

The basic game is the voluntary contribution mechanism (hereafter, VCM), where groups of four subjects are randomly matched in each period and interact for 10 periods. In each period, the subjects are endowed with 20 tokens. Each subject allocates the tokens either to the private account or to the public account. Tokens allocated to the private account remain the same. Each token contributed to the public account yields 1.6 tokens.[note 2] Tokens in the public account are evenly distributed to four group members. Let $c_{it}$ denote the individual’s contribution to the public account in period t, restricted to an integer satisfying $0\leq c_{it}\leq 20$. Let $\pi_{it}$ denote individual *i*’s earnings from their private account and the public account in period t, given the contributions of all group member ${\left\{c_{it}\right\}}_{i=1}^{4}$, as follows:$$\pi_{it}=20-c_{it}+\frac{1.6\times \sum_{j=1}^{4}c_{jt}}{4}$$

The experiment includes a baseline treatment and four distinct leader selection mechanisms as follows:***Control*** **(Baseline Treatment)**. All subjects play the standard *VC*M. They independently and simultaneously choose the contributions to the public account.***RL*** **(Randomly Selected Leadership)**. One individual is randomly selected as the group leader and chooses the contribution first. The followers perfectly observe the leader’s contribution and then contribute simultaneously***VL*** **(Voluntary Leadership)**. Group members first volunteer to be the leaders. All volunteers automatically become leaders. Each leader observes the number of leaders and then simultaneously chooses a contribution. The followers (if there are any) perfectly observe the contributions of the leaders and then simultaneously choose the contributions.***RL-VC*** **(Randomly Selected Leadership from Voluntary Candidates)**. Group members first volunteer to be the leaders. Unlike *VL*, only one volunteer is randomly selected to be the leader. The leader does not know how many volunteers they compete with and choose the contribution. The followers perfectly observe the leader’s contribution and then simultaneously choose the contribution.***VL-RC*** **(Voluntary Leadership by the Randomly Selected Candidate)**. One individual is randomly selected and decides whether to be the leader. If the individual volunteers to be the leader, then, they contributes first, and the rest follow after observing their contribution. If the individual does not want to be the leader, all group members contribute simultaneously.

In all treatments, subjects anonymously receive feedback on group members’ roles, contributions, and payoffs at the end of each period. In stark contrast to the fixed-pairing design of [18] ([18]), subjects are randomly re-matched after each period. In a fixed-pairing design, subjects are paired with the same individuals throughout the experiment. Behaviors in previous periods have a lasting impact on future periods. Such a design is useful for understanding the dynamic interactions of the subjects, how subjects coordinate to some equilibrium profile, and the reputation effect of the leadership selection mechanisms. In contrast, in a random pairing design, subjects are paired with new teammates in each period. Subjects effectively played a series of independent one-shot games, because subjects have no history of play to condition on when devising strategies in each period. Compared with [18] ([18]), our design avoids the concerns of multiple equilibria, typical in repeated games with incomplete information. Our design also eliminates the contamination whereby subject behaviors may condition on the divergent beliefs of other participants’ cooperative type due to history of play. Our design makes the one-shot interaction within each leadership selection mechanism the play’s focus and generates more independent observations than fixed-pairing design.

### 3.2. Treatment Effects

Similarly to what [19] ([19]) finds under a game-theoretic framework in public goods games with leadership, we believe leaders increase the provision of public goods through the behavioral force of signaling and reciprocity. We conjecture that what really differentiates the effectiveness of four leadership mechanisms theoretically is how *informative* the leadership selection mechanism is in communicating the average degree of generosity of group members to followers through the leaders’ (in)actions.

We perceive a leadership selection mechanism as a signaling device. The state of the world is the average degree of generosity of group members. A leadership selection mechanism is a family of distributions of signals conditional on the average degree of generosity. A signal is the contribution level of a leader. Upon observing the contribution levels, followers update their beliefs about the average degree of generosity of their group members. Then, reciprocity kicks in as follows: followers reward cooperation and punish free-riding.

Table 1 summarizes the properties of the four leader selection mechanisms. We first explain the properties and then connect these properties to the signaling power of leadership selection mechanisms.

We first discuss the treatments’ facts regarding the number of leaders. *VL* is the only treatment allowing multiple group leaders (Fact F1). *RL* is an *exogenous* leader selection mechanism and the only treatment where a leader always exists. *VL*, *RL-VC*, and *VL-RC* are *endogenous* leader selection mechanisms; a group member becomes a leader only if he or she is willing to lead, and there could be groups without a leader (Fact F2).

A mechanism *respects individual willingness to lead* (Attribute A1) if any group member willing to lead is selected as a leader. Such a mechanism may have multiple leaders (A1 implying F1). When there are multiple leaders, they have incentives to free-ride on each other (Effect E1).

A mechanism *respects individual willingness not to lead* (Attribute A2) if any group member unwilling to lead is not selected as a leader. Such a mechanism may potentially have no leader (A2 implying F2). By revealed preference, those who volunteered are willing to lead (Effect E2).

A mechanism *respects group willingness to lead* (Attribute A3) if a leader exists when and only when a group member is willing to lead. This attribute distinguishes *VL-RC* from *VL* and *RL-VC*: In *VL-RC*, the group member willing to lead may not be the randomly selected leader candidate. Given that one player’s decision determines the leadership structure in *VL-RC*, we cannot rule out the possibility of other members’ willingness to lead in the leaderless group (Effect E3).

The different effects of the leader selection mechanisms shed light on the channels through which leadership may operate. First, in the absence of the “*free riding incentive among multiple leaders*” (E1), leaders in *RL*, *RL-VC*, and *VL-RC* should contribute more than their counterparts in *VL*. A leader in *VL* might feel that the other leaders will take care of the group and then lower their contribution.

Second, with the “*strong willingness to lead by the leader*” (E2), leaders in *VL*, *RL-VC*, and *VL-RC* should contribute more than their counterparts in *RL*. Randomly selected leaders in *RL* may not be intrinsically motivated to unite the team and set a good example. In contrast, self-motivated leaders in the endogenous leader mechanisms intend to promote group contributions.

Third, with the “*no willingness to lead by group members in case of no leader*” (E3) in effect, group members in *VL* and *RL-VC* should contribute less than their counterparts in the *VL-RC* when there is no leader. In *VL* and *RL-VC*, no leader occurs only when all group members refuse to engage in leadership. The unanimous reluctance strongly signals that group members want to free-ride. Thus, groups without a leader can hardly achieve cooperation.

We hypothesize the rankings of the four leader selection mechanisms in raising public goods contribution. First, the three effects are all positive for the *VL-RC* mechanism, making it the most effective. Although a reluctant leader in *RL* may contribute nothing, observing such inaction does not give information about other members’ inclinations to cooperate. In contrast, no leader in *RL-VC* strongly signals that group members want to free-ride. *RL* suffers the negative effect of E2, and *RL-VC* suffers the negative effect of E3. *RL* and *RL-VC* are intermediately effective. Last, the three effects are all negative for the *VL* mechanism, making it the least effective mechanism.

### 3.3. Procedure

One hundred and sixty college students from Tsinghua University participated online between April and July 2020. We assigned 32 students to each treatment and ran two sessions per treatment. In each session, 16 subjects were randomly assigned to groups of four and re-matched after each period.

Subjects received an electronic version of the instructions and were required to complete the task independently in a quiet place without being disturbed. The experimenter guided the subjects through the details in the instructions by organizing an online meeting and encrypted their true identities to minimize the possibility of any communication between them. Any further questions from the subjects were privately answered.

The experiment was programmed and conducted using the software z-Tree (version 4.1.6) (https://www.ztree.uzh.ch/en.html) ([10]). Each session lasted approximately 20 min. The earnings were accumulated across the 10 periods with the conversion rate of 1 token = CNY 0.05. The average payment received was about CNY 23.597 per subject (including a CNY 5 show-up fee), comparable to standard online payment levels in mainland China.

## 4. Results

In this section, we report our experimental results. First, we compare the average contributions by treatment. Next, we compare the average contributions between groups with and without a leader. Then, we compare the leader–follower interactions by treatment. Lastly, we focus on the free-riding incentives in endogenous leadership selection mechanisms.

### 4.1. Comparison of Average Contributions

**Result 1**. Introducing leadership selection mechanisms did not significantly increase public goods contributions. There was no significant difference in average contributions among *Control*, *VL-RC*, *RL*, and *RL-VC*. *VL* is the least effective in raising contributions.

Table 2 lists the average contributions of all treatments. In none of the leadership treatments did the contribution levels significantly exceed those in the baseline treatment (Wilcoxon rank-sum test, *RL* vs. *Control*: *p* = 0.098; *VL* vs. *Control*: *p* < 0.001; *RL-VC* vs. *Control*: *p* = 0.216; *VL-RC* vs. *Control*: *p* = 0.229). The average contribution in *VL* is significantly lower than that in *Control*, *RL*, *RL-VC*, and *VL-RC*. Further 0.341 (*p* < 0.001), indicating a significant difference between the two. The comparison between *RL* and *VL* shows an effect size oanalysis is provided in Appendix B, Table A1. The effect size between the *Control* and the *VL* is 3.341 (*p* < 0.001), indicating a significant difference between the two. The comparison between *RL* and *VL* shows an effect size of 5.106 (*p* < 0.001, confidence interval [3.672, 6.541]), further demonstrating that the contribution in the RL group is significantly higher than in *VL*. Notably, the comparison between *VL*-RC and *VL* reveals an effect size of −4.619 (*p* < 0.001, confidence interval [−5.995, −3.243]), indicating that the contribution in *VL-RC* is significantly higher than in *VL*. This result highlights the significant advantage in contributions of *VL-RC*, showing a marked improvement over *VL*. The average contributions in *RL* and *VL-RC* are significantly greater than that in *VL* and are similar to or slightly greater than the average contribution in the *Control*.[note 3]

Figure 1 depicts the average contributions over periods by treatment. Contributions in the *VL* are consistently lower than those in the baseline treatment across most periods (Wilcoxon matched-pairs signed-rank test, *VL* vs. *Control*: *p* < 0.001). Besides, the temporal patterns of contributions for the *Control*, *RL*, *RL-VC*, and *VL-RC* are pretty similar (Wilcoxon matched-pairs signed-rank test, *RL* vs. *Control*: *p* = 0.050; *RL-VC* vs. *Control*: *p* = 0.604; *VL-RC* vs. *Control*: *p* = 0.121).

### 4.2. Comparison of Average Contributions Between Groups with and Without a Leader

**Result 2**. The barrier to becoming a leader amplifies the signaling effect of self-chosen leadership in endogenous leadership treatments. For groups without a leader, the average contribution in endogenous leadership treatments is significantly lower than in the baseline treatment. For groups with a leader, *RL-VC* and *VL-RC* outperform *RL*, and *RL* outperforms *VL*.

To understand the gaps in raising public goods contribution among the mechanisms, we first examine the leadership performance by comparing groups with and without leaders, as shown in Table 3. In every endogenous leader selection mechanism (*VL*, *RL-VC*, and *VL-RC*), groups without a leader contribute much less compared to groups with leaders (Wilcoxon rank-sum test, *p* < 0.001 for all pairwise comparisons). Groups without a leader contribute at an average of around 54.2% of the level of groups with leaders, comparable to treatments in the study of [15] ([15])[note 4].

In endogenous leadership treatments, self-chosen leadership serves as a signaling device. When there is no leader, the structure of the subgame is essentially the same as that in the control treatment. The subjects who avoid an early contribution are perceived to be more uncooperative than when a simultaneous move is the only option ([31]). The average contribution in groups without a leader in the three treatments is significantly lower than in the *Control* (Wilcoxon rank-sum test, *p* < 0.001 for all pairwise comparisons, further details can be found in Table A2 and Table A3).

Furthermore, the barrier to becoming a leader matters in fostering group contributions. *VL-RC* only allows one group member to be a leader. When the randomly selected candidate refuses to lead, subjects are still not sure about the followers’ willingness to cooperate. In contrast, *VL* and *RL-VC* allow multiple group members to be candidates. When there is no leader, group members perceive others as not cooperative. We find average group contribution when no leader exists in *VL-RC* is significantly higher than that in *VL* or in *RL-VC*. (Wilcoxon rank-sum test for the group without a leader, *VL* vs. *RL-VC*: *p* = 0.169; *VL* vs. *VL-RC*: *p* < 0.001; *RL-VC* vs. *VL-RC*: *p* = 0.026).

Restricting attention to groups with a leader, we compare the average group contributions of the endogenous leader selection mechanisms against that of *RL*. We find that *VL-RC* outperforms *RL*, while *VL* performs significantly worse (Wilcoxon rank-sum test for groups with a leader, *RL* vs. *VL-RC*: *p* = 0.002; *RL* vs. *VL*: *p* < 0.001; *RL* vs. *RL-VC*: *p* = 0.651). Among the three endogenous leadership treatments, *VL-RC* significantly outperforms *RL-VC* and *VL*; *RL-VC* significantly outperforms *VL* (Wilcoxon rank-sum test, *p* < 0.001 for all pairwise comparisons).

In all three endogenous leader selection mechanisms, we find “*reluctance to lead*”. The average vacancy rate of voluntary leadership is around 40%. The leader’s presence is the greatest in *RL-VC* and the lowest in *VL-RC*. On the other hand, more groups with leaders do not always translate to greater average contributions. *VL* allows multiple leaders, leading to a diffusion of responsibility among them, which could limit the increase in contributions.

### 4.3. Comparison of Leader–Follower Interactions

**Result 3**. Leaders lead by example, and followers follow. Within the endogenous leader selection mechanisms, the leader’s influence is more pronounced if the barrier to becoming a leader is lower. There is a significant positive correlation between leader and follower contributions in *VL* and in *RL-VC*, and a weak or no correlation in the other treatments.

In this subsection, we seek to understand the leader’s role modeling effect (or lack thereof) by examining leaders’ cooperative behavior and followers’ reciprocity.[note 5] In aggregate, leaders contribute more and effectively foster greater follower contribution. Table 4 presents the average contributions by role at the individual level, conditional on having group leaders. Regardless of treatment, leaders contribute significantly more than followers (Wilcoxon rank-sum test, *p* < 0.05 for all pairwise comparisons, further details can be found in Table A4 and Table A5).[note 6]

Table 5 presents the average follower contributions in endogenous leader selection mechanisms. The contribution levels of followers in groups with a leader are significantly greater than those in groups without a leader (Wilcoxon rank-sum test, *p* < 0.001 for all pairwise comparisons, further details can be found in Table A5 and Table A6).

Failure to become a leader dampens volunteers’ willingness to contribute. The average contribution of the volunteers who failed to become a leader in *RL-VC* (mean = 11.136) is between that of the leaders (mean = 12.667) and that of the members who did not volunteer (mean = 3.861).

The barrier to becoming a leader also matters in coordinating leader–follower contributions. Figure 2 depicts the average contributions over periods made by leaders and followers at the group level and the average contribution of groups without a leader. Spearman’s rank correlation analyses reveal mixed results, with a significant positive relationship observed in the *VL* and in *RL-VC*, while other treatments show weak or no significant correlation between leader and follower contributions.[note 7]

In *VL*, all volunteers become leaders. Observing more leaders making greater contributions gives followers a clear signal of cooperative play. A cooperative, forward-looking leader is confident that the followers will reciprocate the favor. *RL-VC* is a weakened version where followers observe the contribution level of only one leader.

In *RL*, how much the leader contributes does not generate much information about how likely other followers will contribute under our random-pairing design. In *VL-RC*, barriers to becoming a leader are higher than in *VL*, and it is more difficult to infer other followers’ inclinations to cooperate.

We observe no significant difference between the contributions of leaders in periods 1–5 and 6–10, implying that leaders exhibit stable pro-social behavior ([30]). Followers in *RL* and in *RL-VC* exhibit a noticeable decline in contributions over time. Followers in other leadership treatments show no significant variation in their contributions. Members of groups without a leader in *VL*, *RL-VC*, and *VL-RC* significantly decrease their contributions over periods.[note 8]

### 4.4. Discussion of Endogenous Leadership

**Result 4.** Endogenous leader selection mechanisms that allow multiple volunteers induce more selfish leaders. When there are multiple leaders, the leader with the lowest contribution significantly influences the followers.

Among endogenous leader selection mechanisms, only *VL* and *RL-VC* allow multiple candidates. Followers have incentives to free-ride on other volunteers. However, Table 6 shows the difference in probabilities to volunteer between the two mechanisms (Wilcoxon rank-sum test, *p* < 0.001).

Figure 3 also shows that the number of volunteers in *RL-VC* is higher than in *VL* for most of the experiment. In both treatments, individuals’ volunteer tendencies fluctuate considerably, gradually decreasing. The average number of volunteers in *RL-VC* starts above that of *VL* and ends higher.

We attribute the difference to another layer of bystander effect induced by the probability of becoming a leader conditional on being a volunteer. As we highlighted in the second row of Table 6, conditional on being a volunteer in *VL*, candidates are forced to be a leader; on the other hand, conditional on being a volunteer in *RL-VC*, candidates are randomly selected to be the single leader. Thus, in *VL*, volunteers have incentives to free-ride on other leaders, which dampens the incentive to become a volunteer in the first place.

We further find that mechanisms allowing multiple candidates (*VL* and *RL-VC*) induce more selfish leaders than mechanisms allowing only one candidate (*VL-RC*). Conditional on being a leader, the share of the leaders contributing no more than one-quarter of their endowment is 50% in *VL*, 28.33% in *RL-VC*, and only 7.69% in *VL-RC*, though the share of selfish leaders in the exogenous leader selection mechanism of *RL* is 31.25%.

When multiple leaders emerge, followers base their contribution levels on those made by the worst leader to a large degree. In *VL*, Spearman’s rank correlation coefficient is 0.283 (*p* = 0.063) with the best leader contribution, and the coefficient is 0.419 (*p* = 0.005) with the worst leader contribution.

## 5. Conclusions

We study leader selection mechanisms in the public goods provision game. Our analysis yields four key findings. First, introducing leadership selection mechanisms does not increase contributions; voluntary leadership is the least effective and performs worse than the baseline. Second, the presence of a leader significantly increases overall contributions, and mechanisms with higher barriers (e.g., *VL-RC*) outperform those with lower barriers (e.g., *VL*). Third, leaders lead by example, and followers reciprocate, but the strength of this relationship depends on the leadership mechanism. Finally, mechanisms that allow multiple leaders (e.g., *VL* and *RL-VC*) tend to induce more selfish behavior, as the diffusion of responsibility among leaders weakens their influence.

Our study reinforced the finding of [18] ([18]) that voluntary leadership from a randomly selected candidate (*VL-RC*) is a promising mechanism for raising public goods contribution. Compared with [18] ([18]), we report average contributions by treatment and justify our experimental design through the behavioral theory of signaling and reciprocity. *VL-RC* enjoys the following three good properties: the leader, by revealed preference, is cooperative; the leader has no free-riding incentives; and no leader leaves members hoping that someone cooperative is not lucky enough to be selected, increasing confidence in cooperative play. Our random-pairing design makes the subjects more focused on the one-shot game structure, highlighting the signaling effect of leadership, and reducing the concerns with multiple equilibria in repeated games with incomplete information.

Our results suggest that while leadership can enhance public goods contributions, collective leadership is not as ideal as individual leadership. Mechanisms with higher barriers to leadership are more effective in fostering cooperation, as they reduce free-riding incentives and strengthen the signaling effect of leadership. In designing the leadership selection mechanism, policymakers and organizational leaders can create barriers in selecting candidates and privately encourage the candidate who values the common good the most to lead by example voluntarily.

Although our leadership selection mechanisms are motivated by real-world examples such as fundraising and climate policies, we draw our findings from a controlled lab setting using college students as research subjects with small-stake decisions. We only study a simple form of public goods game where leaders are of equal status, and group members cannot communicate with each other. In future research, it would be interesting to see if our findings can be generalized to individuals with heterogeneous social-economic backgrounds, who are called upon to make high-stake decisions. It would also be interesting to see if a more complicated and more real-world relevant form of public goods game with multi-layered leadership hierarchies and communication between subjects would point to a leadership selection mechanism with similar properties ([17]; [23]; [4]).

## Figures and Tables

**Figure 1 behavsci-15-00444-f001:**
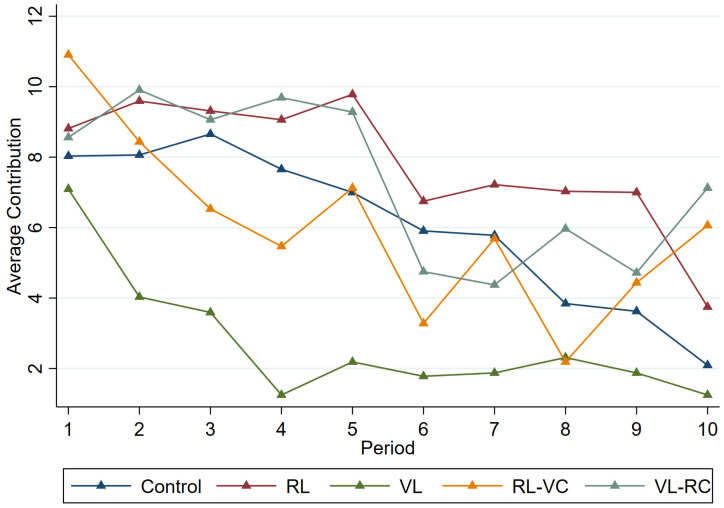
Average contributions over periods by treatment.

**Figure 2 behavsci-15-00444-f002:**
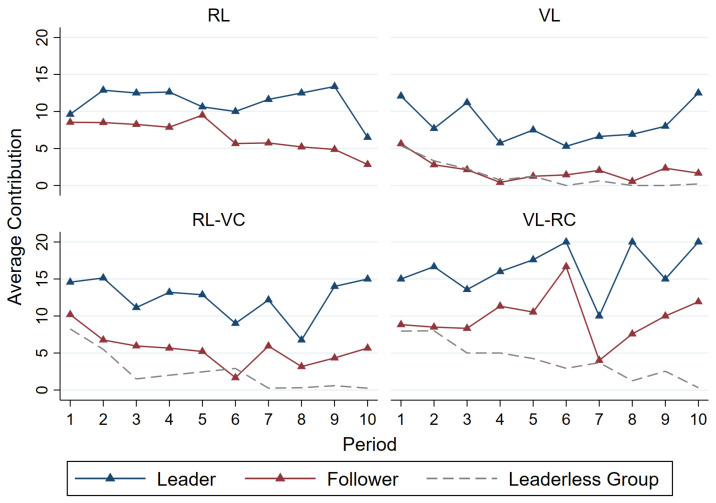
Average contribution over periods by role.

**Figure 3 behavsci-15-00444-f003:**
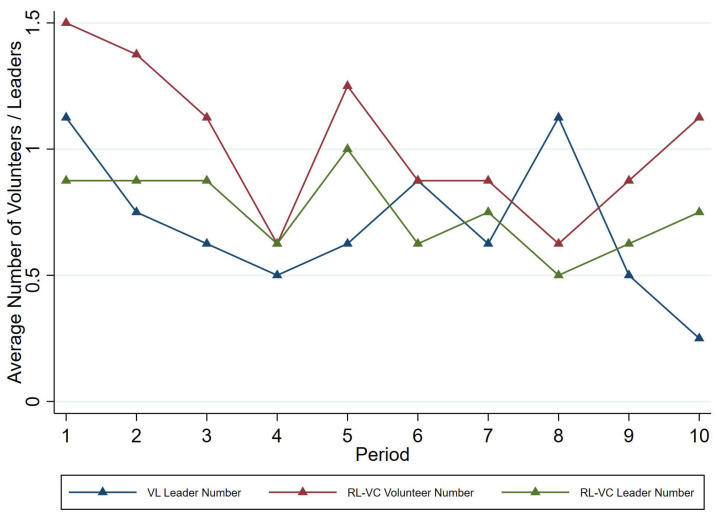
Average number of volunteers or leaders over periods by treatment.

**Table 1 behavsci-15-00444-t001:** Summary of differences in the leader selection mechanisms.

	*RL*	*VL*	*RL-VC*	*VL-RC*
**F1. Possibility of multiple leaders**	No	Yes	No	No
**F2. Possibility of no leader**	No	Yes	Yes	Yes
**A1. Respecting individual willingness to lead**	No	Yes	No	No
**A2. Respecting individual willingness not to lead**	No	Yes	Yes	Yes
**A3. Respecting group willingness to lead**	—	Yes	Yes	No
**E1. Free riding incentive among multiple leaders**	No	Yes	No	No
**E2. Strong willingness to lead by the leader**	No	Yes	Yes	Yes
**E3. No willingness to lead in case of no leader**	—	Yes	Yes	No

**Table 2 behavsci-15-00444-t002:** Average contribution separated by treatment.

	*Control*	*RL*	*VL*	*RL-VC*	*VL-RC*
**Contribution**	6.066	7.831	2.725	6.013	7.344
	(3.587)	(5.700)	(3.220)	(5.498)	(5.392)

Notes: The unit of observation is the group. Standard deviations are in parentheses.

**Table 3 behavsci-15-00444-t003:** Average contribution separated by group leadership.

	*Control*	*RL*	*VL*	*RL-VC*	*VL-RC*
**Groups without a leader**	6.066	—	1.121	1.700	3.829
	(3.587)		(1.899)	(2.335)	(3.425)
**Groups with a leader**	—	7.831	3.972	7.450	11.038
		(5.700)	(3.493)	(5.506)	(4.555)
**Percentage of groups with a leader (%)**	0	100	56.25	75	48.75

Notes: The unit of observation is the group. Standard deviations are in parentheses.

**Table 4 behavsci-15-00444-t004:** Average contributions by role in groups with a leader.

	*RL*	*VL*	*RL-VC*	*VL-RC*
Leader	11.225	8.214	12.667	16.231
	(7.286)	(6.412)	(7.191)	(5.905)
Follower	6.7	2.056	5.711	9.308
	(6.999)	(3.783)	(6.974)	(7.672)

Notes: The unit of observation is the individual. Standard deviations are in parentheses.

**Table 5 behavsci-15-00444-t005:** Average follower contributions in endogenous leader selection mechanisms.

	*VL*	*RL-VC*	*VL-RC*
Has a leader	2.056	5.711	9.308
	(3.783)	(6.974)	(7.672)
No leader	1.121	1.700	3.829
	(3.370)	(3.584)	(6.299)

Notes: The unit of observation is the individual. Standard deviations are in parentheses.

**Table 6 behavsci-15-00444-t006:** Inclinations to volunteer and lead.

	*VL*	*RL-VC*	*VL-RC*
Probability of being chosen as a candidate (%)	100	100	25
	by design	by design	by design
Conditioning on being a candidate,	17.5	25.625	48.750
probability to volunteer (%)	(0.381)	(0.190)	(0.503)
Conditioning on being a volunteer,	100	73.171	100
probability to lead (%)	by design	(0.446)	by design
Conditioning on being a candidate,	17.5	18.75	48.75
probability to lead (%)	(0.381)	(0.391)	(0.503)
Unconditional probability of being a leader (%)	17.5	18.75	12.188
	(0.381)	(0.391)	(0.328)

Notes: The unit of observation is the individual. Standard deviations are in parentheses.

## Data Availability

The datasets used and analyzed in this study are available on Mendeley Data. DOI: 10.17632/85p9mh3zmp.2.

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
