# Peer review of "How to Select the Leader in a One-Shot Public Goods Game: Evidence from the Laboratory"

_behavsci, 2025, doi:10.3390/bs15040444_

Round 1
Reviewer 1 Report
Comments and Suggestions for Authors
Your manuscript provides a thorough and thoughtful examination of leadership selection mechanisms in public goods settings. The logical design, clear data analysis, and ties to real-world applications are definite strengths. With just a bit more discussion on how random matching influences behavior and any limitations to broader applicability, the paper will be even more robust. Overall, it’s a well-executed study that adds meaningful insights to the literature. Here are some minor recommendations for improvement:
-
Strengthening Theoretical Context (Section 2, approx. page 2, paragraphs 2–4)
Your Literature Review effectively references major works (e.g., Hermalin 1998; Rivas and Sutter 2011). However, incorporating a short but explicit explanation of how each leadership mechanism (e.g., RL, VL, RL-VC, VL-RC) connects to core theories of signaling and reciprocity would help readers fully grasp the rationale behind your hypotheses. This additional framing can sharpen the reader’s understanding of why certain mechanisms might yield higher contributions. -
Clarifying Random Matching Logic (Section 3, approx. page 4, paragraph 2, and page 5, paragraph 1)
You note that subjects are randomly rematched each period, which helps avoid long-term targeted reciprocity. A brief expansion on the significance of this design choice—versus a fixed-pairing or partner design—would clarify why each new period resembles a fresh, almost “one-shot” game. Readers unfamiliar with experimental economics will benefit from seeing exactly how random rematching influences equilibrium expectations or participants’ strategies. -
Expanding Discussion of External Validity (Section 5, near final paragraphs on page 10)
Although you draw parallels to real-world scenarios (e.g., charity fundraising, climate policy), adding a concise paragraph about how and when these lab findings might (or might not) generalize to larger, more diverse groups or to higher-stakes environments could strengthen your conclusion. Readers often wonder whether small-group student samples translate to field settings. Explicitly acknowledging the scope of these results underscores their usefulness and any potential limitations. -
Enriching Statistical Reporting (Section 4, roughly pages 7–9)
Tables and graphs are clear, and your use of rank-sum tests and correlation measures is appropriate. To give readers a deeper sense of how large or meaningful the observed differences are, consider presenting effect sizes or confidence intervals alongside p-values. This would not only emphasize statistical significance, but also illuminate practical or theoretical significance. -
Minor Points for Clarity and Flow (general reading of pages 3–4)
Overall, the paper’s English is well-structured. That said, re-check transitions between sections—especially when moving from the Research Design to the Results—to ensure each subsection connects seamlessly. Minor editorial refinements (e.g., consistent tense usage, clarifying pronouns, or explicitly labeling each subsection) will make the text even more reader-friendly. -
Implications and Future Directions (Section 5, approx. page 10, paragraphs 2–3)
Your conclusion that voluntary leadership from a randomly selected candidate (VL-RC) outperforms the other mechanisms is compelling. Expanding on how organizations or policy-makers might adopt such a mechanism in real-world collaboration—along with any conditions necessary for success (e.g., communication channels, repeated interactions)—would be valuable. You might also consider briefly suggesting further studies on hierarchical or multi-level leadership structures, which could extend your analysis beyond the scope of a single leader or single-shot decision-making.
Author Response
Thank you for your very helpful comments. Please see the attached response letter for details.

Reviewer 2 Report
Comments and Suggestions for Authors
This paper is an experiment, very nicely done and presented. I must say I liked the paper; it is indeed a nice piece of work overall.
The experiment considers 4 different treatments of leadership (other than the base line) in a contribution to public good game.
The set-up is excellent. My comments are therefore minor that might help the authors to present the findings in a better way.
First, in the payoff, there should be a discussion on the parameters chosen for the game. For example, why 1.6? why 20? These chosen numbers must be justified, also compared with the literature.
The paper is very close to He and Zheng (2024); however the results have not been compared directly. The authors should be elaborate, critical and honest here!
The literature review is good but not very informative. I think it should be placed later, perhaps in the concluding section and should contrast the results found here.
I didn’t find any specific typos; however the author must do an overall good editorial check for any possible “mistakes” in presentation, usage of grammar and in formatting.
To sum up, I do think this paper has presented a nice experiment and should be accepted for publication; however, the authors should revise it with an aim to place it within the existing literature more carefully.
Author Response

(The authors gave the same response as above.)

Reviewer 3 Report
Comments and Suggestions for Authors
Review of “How to Select the Leader in a One-Shot Public Goods Game: Evidence from the Laboratory”
Summary
The authors present results from simple, one-shot linear public goods games. All games involve four symmetric players who start with 20 tokens that they can keep or contribute to the public good. The public good has a social return of 1.6 tokens per token invested but a private return of 0.4 tokens per token invested, making the Nash equilibrium to contribute nothing. The authors compare the standard linear public goods game in which all four players choose their contributions simultaneously (the control treatment) to four additional treatments (VL, RL, RL-VC, and VL-RC), allowing different forms of leadership. Here, leadership is defined in terms of sequence and information: the leader makes his/her contribution prior to the other players in the game, who are informed of the leader’s contribution before the other players make their own contributions.
The four treatments involve different ways of selecting the leader. In the VL treatment, any player (or no player) may volunteer, and all players who volunteer are leaders. In the RL treatment, one player is randomly assigned to be the leader, and may not refuse to be the leader. In the RL-VC treatment, as many players as wish may volunteer to be the leader, but only one player is assigned (randomly) to actually be the leader. In the VL-RC treatment, one player is randomly selected to be the leader, but that player has the option to turn down the position, leaving the group leaderless.
The authors find that the VL treatment generates the lowest amount of contributions, even lower than the control treatment in which there are no leaders. The other three leadership treatments do not generate statistically different levels of contributions (although RL and VL-RC do generate slightly higher levels of contributions). The authors show that in treatments that allow for voluntary leadership, the existence of that leadership matters – groups in which no one volunteers to be the leader have statistically significantly lower levels of contributions. Finally, the researchers find that leaders contribute more than followers and that followers only positively respond to leaders’ contributions in the VL treatment.
Analysis
I like this paper very much and recommend that it be published with minimal revisions. Some of the strengths of the paper are:
- The authors have chosen an interesting topic – endogenous leadership – with significant real-world applications. The authors mention the usefulness of voluntary leadership from celebrities in fundraising campaigns, but I believe these findings apply to many more situations. I think anytime you have the potential for a group to form to fund a public good, leadership will impact that group's success, and this research speaks to that process.
- The authors fit their research into the broader leadership in public goods provision literature. Their paper fills a gap in that research and adds to our knowledge. Their overall literature review seems complete and comprehensive.
- The authors’ experimental design is solid. They use a common public goods game, and their treatments inducing leadership seem effective. The instructions are provided, and are clearly written. Subjects were given reasonably strong incentives. The one-shot, anonymous re-matching protocol is a good way to measure the hypothesized effects.
- Their data analysis is clear. The authors mainly stick to estimating effects that correspond to their treatments, and present basic data in tables, graphs, and with the appropriate statistical analysis.
- The paper is well-written, clearly organized, and not too long!
Suggestions for Improvement
I think the paper could be improved in several ways:
- Upon my initial reading of the paper, I was confused as to how the authors were using the term leadership. It took me until the Methods section to really see that this meant first movement in a public goods game. Leadership is a broad academic subject, with many potential definitions, and so I would suggest that the authors clearly indicate in the introduction how they are defining leadership – as the first move in a sequential public goods game. This would also make it easier to understand the literature review.
- Given the narrow definition of leadership, I think the literature review could be a bit clearer. In particular, it would be helpful to clearly indicate what types of leadership each paper is measuring (if it is not already clear).
- In section 3.2, Treatment Effects, the authors provide some explanations as to how they expect their results to come out. Although the explanations seemed reasonable, I would have liked a stronger link to previous research on public goods and behavioral theory. The authors seem to take it as given that one player’s contributions could influence another’s, but they don’t really back up this claim with reference to empirical or theoretical work. The most straightforward analysis of this game, using classical game theory, would predict zero contributions in every treatment. Models incorporating other-regarding preferences will generate different predictions under these treatments, but the specifics are not discussed. For example, inequity aversion would seem to imply that if the leader contributes a positive amount, the follower would possibly want to contribute as well to reduce the inequity between leader and follower. The authors don’t need to generate a full theoretical model, but I would like some more detail about their assumptions about other-regarding behavior and explanation as to how these assumptions generate predictions of positive contributions to the public good.
- From the text, I see that there were 160 total participants across 5 treatments. I assume that means there were 32 participants in each treatment? Please clarify the distribution of participants across treatments in the paper’s text.
- The authors indicate that the data is available upon request. I request that they make the data available in a public repository like Mendeley. For me, this is the only non-negotiable recommendation – I will not recommend publication unless the data is in a public repository.
Author Response

(The authors gave the same response as above.)
